# In Vivo and In Vitro Experimental Study Comparing the Effect of a Combination of Sodium Dichloroacetate and Valproic Acid with That of Temozolomide on Adult Glioblastoma

**DOI:** 10.3390/ijms26146784

**Published:** 2025-07-15

**Authors:** Rūta Skredėnienė, Donatas Stakišaitis, Angelija Valančiūtė, Ingrida Balnytė

**Affiliations:** 1Department of Histology and Embryology, Medical Academy, Lithuanian University of Health Sciences, 44307 Kaunas, Lithuania; ruta.skredeniene@lsmu.lt (R.S.); angelija.valanciute@lsmu.lt (A.V.); ingrida.balnyte@lsmu.lt (I.B.); 2Laboratory of Molecular Oncology, National Cancer Institute, 08660 Vilnius, Lithuania

**Keywords:** glioblastoma, dichloroacetate-valproic acid combination, temozolomide, cancer markers, chicken embryo chorioallantoic membrane model

## Abstract

To date, there is no effective treatment for glioblastoma (GBM). This study aimed to compare the effectiveness of sodium dichloroacetate (NaDCA), a valproic acid and NaDCA combination (VPA–NaDCA), or temozolomide (TMZ) on U87 and T98G cell tumors on the chick embryo chorioallantoic membrane (CAM), and on the expression of proliferating cell nuclear antigen (PCNA), polycomb inhibitory complex catalytic subunit 2 (EZH2), and *TP53* gene-encoded p53 protein (p53) in tumors on the CAM, and *SLC12A2* (gene encoding Na^+^-K^+^-2Cl^−^ (NKCC1) co-tarnsporter), *SLC12A5* (gene encoding K^+^-Cl^−^ (KCC2) co-transporter), *SLC5A8* (gene encoding Na^+^-dependent monocarboxylate transporter) and *CDH1* (gene encoding the E-cadherin protein) and *CDH2* (gene encoding the N-cadherin protein) in cells. VPA–NaDCA and TMZ reduced the invasion of U87 and T98G tumors, as well as the expression of PCNA and EZH2 in the tumor. TMZ reduced p53 expression in tumors from both cell lines, whereas VPA–NaDCA did not affect the expression of this marker. VPA–NaDCA, but not TMZ, reduced *SLC12A2* expression in T98G cells. However, VPA–NaDCA and TMZ did not affect *SLC12A2* expression in U87 cells. VPA–NaDCA increased *SLC5A8* expression only in U87 cells, and TMZ did not affect gene expression in either cell line. Only VPA–NaDCA increased *CDH1* expression and decreased *CDH2* expression in T98G cells, whereas TMZ had no effect on gene expression in the study cells. This study demonstrated that VPA–NaDCA exhibits a more effective anticancer effect than NaDCA. The data suggest that VPA–NaDCA has a more effective impact than TMZ; however, the effect of investigational medicines on carcinogenesis varies depending on the cell line. The study of the efficacy of drugs used to treat tumors on the CAM and cells demonstrates that it is essential to assess the effectiveness of treatment, which should be personalized, before administering chemotherapy.

## 1. Introduction

Glioblastoma (GBM) can occur at any age. The disease is more common in the elderly [1,2], is more frequently diagnosed in men [3], and there are known sex-related differences in survival time after GBM diagnosis [4,5].

Temozolomide (TMZ) has been used to treat newly diagnosed GBM since 2005 [6,7]. In treating GBM after surgery, the addition of TMZ to radiotherapy has increased overall survival (OS) by 2 months [8], with a median OS of 15 months and a progression-free survival of 6 months [9,10]. In over 50% of GBM patients, TMZ treatment is ineffective.

GBM cells are characterized by suppressed apoptosis and cell death [11]. The main effect of drugs on GBM is associated with inhibiting tumor cell proliferation and activating apoptosis [12]. Activated glycolytic processes characterize GBM cells [13]. One of today’s targets for treating GBM is the inhibition of pyruvate dehydrogenase kinase (PDK) activity, which results in reduced glycolytic activity associated with carcinogenesis [14]. PDK expression is activated in tumor cells [15,16].

It is essential to find ways to reduce the resistance of GBM cells to treatment. Sodium dichloroacetate (NaDCA) may increase the sensitivity of GBM cells to TMZ [17,18]. Studies in the Balb/c nude mouse model showed that NaDCA monotherapy inhibited tumor growth of GBM U87 cells in the subcutaneous tissue [19]. Research on the U87 tumor in the brain of immunodeficient mice showed that the combination of NaDCA and radiotherapy increased the survival of mice [20]. NaDCA has an inhibitory effect on the growth of GMB cell spheroids [21].

One of the most common adverse reactions to NaDCAs in patients is reversible neuropathy [22]. Research findings support the view that sodium valproate (VPA) can be used to treat neuropathies in cancer patients [23,24]. VPA is a histone deacetylase (HDAC) inhibitor that inhibits the development of chemotherapy-related neuropathy. HDAC inhibition increases neuronal survival through mitophagy and autophagy mechanisms, while maintaining neuronal metabolism intact [25,26]. VPA, an epigenetic modulator, can affect gene expression, inhibit cell proliferation and the cell cycle, and induce apoptosis via demethylation mechanisms [27,28]. Three specific HDAC inhibitor drugs, vorinostat, romidepsin, and VPA, have been investigated in pre-clinical and clinical trials to elucidate their effects on GBM [29,30]. VPA may improve the intracellular delivery of dichloroacetate anions via mitochondrial mechanisms [31,32]. The synergistic effect of chemotherapy and HDAC inhibitors is promising for treating GBM to avoid chemotherapy resistance [33]. VPA alone or combined with other therapies inhibits glioma growth in vivo and in vitro [34]. Non-toxic doses of VPA increase the sensitivity of U87 and T98G cells to gefitinib by inhibiting cell growth through the activation of autophagy [35]. VPA acts by binding E2F transcription factor 1 to the glycosylated GPI and PGK1 promoter. This effect of VPA on glycolysis suggests a new therapeutic strategy in the search for VPA effects on tumor cells [36].

The inflammatory microenvironment of the GBM tumor, characterized by the release of pro-inflammatory cytokines and chemokines, as well as the activation of inflammatory pathways, promotes tumor malignancy [37]. The tissue microenvironments of cancer and inflammatory diseases share similarities: they are characterized by tissue hypoxia, accompanied by elevated lactic acid levels [38]. A common feature of GBM is tissue necrosis accompanied by inflammatory changes in the tumor microenvironment. In combination with necrosis, immunosuppressive inflammation is associated with higher resistance of GBM to treatment [39]. NaDCA is an investigational anti-tumor drug and, due to its properties, also has an anti-inflammatory effect [40]. VPA’s immunomodulatory and anti-inflammatory effects are also known [41,42]. A valproic acid and NaDCA combination (VPA–NaDCA) treatment increased the expression of *Slc5a8* in mice thymocytes, indicating an effect of VPA on the dichloroacetate anion carrier, and significantly affected the expression of genes related to inflammation and the immune response, suggesting that VPA–NaDCA exerts an anti-inflammatory effect by inhibiting the pro-inflammatory mechanisms in the thymocytes of male mice, i.e., by suppressing the expression of genes involved in the inflammation and by inducing the expression of genes for the production of anti-inflammatory chemokines and cytokines [43].

The progress of individualized GBM treatment is closely tied to standard treatment, with new treatment strategies being explored, such as chemotherapy using drugs tailored to the molecular and metabolic characteristics of the tumor [44,45]. Understanding the mechanisms of drugs and the development of treatment resistance is crucial for enhancing the efficacy of GBM treatment by evaluating the impact of medications on changes in cancer prognostic markers [46,47].

Researchers point to the chick chorioallantoic membrane (CAM) model as a promising approach for in vivo efficacy studies. It can be used as a 3D tumor treatment model to evaluate anticancer drugs before administering them to the patient [48]. Xenografts formed from GBM cells in the CAM model provide an opportunity to assess the potential efficacy of individual chemotherapy treatment [49].

The purpose of studying the VPA and NaDCA combination is to determine whether it has a synergistic effect or results in a reduction in the dose of the medicines. The experimental studies were designed to determine differences in the impact of treatment with the investigational drugs; determine differences in the response of GBM U87 and T98G cells and their tumors on the CAM; evaluate the effect of NaDCA or the VPA–NaDCA combination on the cancer markers proliferating cell nuclear antigen (PCNA), polycomb inhibitory complex catalytic subunit 2 (EZH2), and *TP53* gene-encoded p53 protein (p53) expression in tumors on the CAM; evaluate the expression of the carcinogenesis-associated co-transporters NKCC1, KCC2, SLC5A8, and E-cadherin and N-cadherin genes; and compare the data with the effects of TMZ.

## 2. Results

### 2.1. Stereomicroscopic and Histologic Images of U87 and T98G Tumors In Vivo on the CAM, the Tumor Ex Ovo, and H-E Histological Images

Figure 1 shows the biomicroscopy images of the U87 and T98G tumors on the chick embryo chorioallantoic membrane (CAM) in the tested groups at the day 9 of embryo development (EDD9) and EDD12, the ex ovo excised tumor with the CAM, and the tumors’ hematoxylin and eosin (H-E) histological images.

In the stereomicroscope image, the control U87 tumor is visually smaller at EDD12 than at EDD9, as the tumor had invaded the CAM mesenchyme by day 12, and part of the tumor is located on the CAM. In the ex ovo image of the control tumor, the vascular network (a “spoked wheel”) is visible. Compared to the control, treatment with 3 mM NaDCA, 50 µM TMZ, and 2 mM VPA–3 mM NaDCA inhibited tumor growth and the formation of a vascular network around the tumor. Compared to U87-control EDD12 in vivo and ex ovo, the respective tumors of U87-3 mM NaDCA, U87-50 µM TMZ, and U87-2 VPA–3 mM NaDCA are visually larger as they are located on the surface of the CAM, the “spoked wheel” is not expressed, and H-E histology shows that the membrane chorionic epithelium is intact.

The stereomicroscopy image of the T98G control tumor shows precise contours in EDD9, while in EDD12, the tumor is reduced in size and expression due to partial invasion into the CAM, the ex ovo image shows a prominent “spoked wheel”, and the H-E preparation shows disrupted chorionic epithelial integrity and invasion of the tumor into the CAM. In vivo and ex ovo images of T98G-3 mM NaDCA and T98G-2 mM VPA–3 mM NaDCA tumors show a vascular network forming around the tumor; the H-E slides show disrupted integrity of the membrane chorionic epithelium and the CAM is thickened under the tumor. Treatment with 50 µM temozolomide (TMZ) inhibited the growth of the T98G tumor: in EDD9 and EDD12, the tumor grows on the CAM; ex ovo, the vascular network around the tumor is absent, and the H-E image shows that the TMZ-treated tumor did not damage the membrane chorionic epithelium integrity.

### 2.2. U87 and T98G Tumor Growth, Rate of Tumor Invasion into the CAM

Table A1 and Figure 2 show the invasion rates of U87 and T98G tumors on the CAM in the study groups.

Compared to the U87-control, treatment with 3 mM NaDCA, 50 µM TMZ, or 2 mM VPA–3 mM NaDCA significantly reduced the incidence of the U87 tumor’s invasion into the CAM. Compared to the T98G tumor control, only treatment with 50 µM TMZ significantly reduced the incidence of invasion. No effect on T98G tumor invasion was found in treatment with 3 mM NaDCA or 2 mM VPA–3 mM NaDCA.

### 2.3. The IMP Impact on Neo-Angiogenesis in the CAM and on CAM Thickness Under the Tumor

Figure 3 shows the fluorescent stereomicroscopic images of U87 and T98G tumors on the CAM in the tested groups at EDD12.

Dextran highlighted the vascular network formed around U87 and T98G tumors. Compared to the U87-control, images of U87-3 mM NaDCA, U87-50 µM TMZ, and U87-2 mM VPA–3 mM NaDCA tumors showed a reduced vascular network (blunted neo-angiogenesis). Compared to the T98G-control, the vascular network around the tumor is reduced in T98G-3 mM NaDCA, T98G-50 µM TMZ, and T98G-2 mM VPA–3 mM NaDCA tumor images.

Table A2 and Figure 4 show the histomorphometric data for the number of vessels in the CAM under the studied U87 and T98G tumors.

Compared to the U87-control, 3 mM NaDCA or 50 µM TMZ and 2 mM VPA–3 mM NaDCA significantly reduced the number of blood vessels in the CAM. The effect of 2 mM VPA–3 mM NaDCA treatment on neo-angiogenesis was considerably more significant than that of 3 mM NaDCA and 50 µM TMZ in the U87 tumor. Treatment with 3 mM NaDCA did not affect the number of blood vessels in the CAM under the T98G tumor. Neo-angiogenesis was significantly blunted by treatment in the T98G-50 µM TMZ and T98G-2 mM VPA–3 mM NaDCA groups. Compared to T98G-3 mM NaDCA, treatment with 50 µM TMZ or T98G-2 mM VPA-3 mM NaDCA significantly reduced the number of vessels in the CAM, with an equal effect on the latter.

Table A3 and Figure 5 show the histomorphometric data on the sub-tumor CAM thickness in the U87 and T98G tumors.

Treatment with 3 mM NaDCA significantly reduced the CAM thickness under the U87 tumor. The CAM thickness under the tumor was significantly lower in the T98G-2 mM VPA–3 mM NaDCA group than in the T98G-control.

### 2.4. The Data of Immunohistochemical Examination of PCNA, p53, and EZH2 Marker Expression in the Tumors

The data for the proliferating cell nuclear antigen (PCNA), *TP53* gene-encoded p53 protein (p53), and polycomb inhibitory complex catalytic subunit 2 (EZH2) markers tested in the tumor are presented in Table A4, Table A5 and Table A6 in Appendix A.

#### 2.4.1. The PCNA Expression of U87 and T98G in Control and Treated Tumors

Table A4 and Figure 6 show the PCNA expression data for the U87 and T98G tumor groups.

The frequency of PCNA-positive cells was similar to that of U87 and T98G control tumors. Compared to U87-control, U87-3 mM NaDCA, and U87-2 mM VPA–3 mM NaDCA tumors, there were no significant differences in PCNA expression, and treatment with 50 µM TMZ significantly reduced the number of PCNA-positive cells in the U87 tumor.

The frequency of PCNA-positive cells was significantly lower in T98G tumors treated with 50 µM TMZ or 2 mM VPA–3 mM NaDCA compared to the T98G-control, and treatment with 3 mM NaDCA did not affect PCNA expression.

#### 2.4.2. The p53 Expression of U87 and T98G in Control and Treated Tumors

Table A5 and Figure 7 show the p53 expression data for the U87 and T98G tumor groups.

No significant differences in p53 expression were found when comparing the U87-control, U87-3 mM NaDCA, and U87-2 mM VPA–3 mM NaDCA groups. However, treatment with 50 µM TMZ significantly reduced the number of p53-positive cells in the U87 tumor, and this number was considerably lower than in the U87-2 mM VPA–3 mM NaDCA group.

The number of p53-positive cells was significantly lower in the tumors treated with 50 µM TMZ compared with the T98G-control, and treatment with 3 mM NaDCA or 2 mM VPA–3 mM NaDCA did not affect the expression of p53 in the T98G tumor compared to the control. The expression of p53-positive T98G cells in the tumor did not differ between the TMZ-treated and 2 mM VPA–3 mM NaDCA groups.

#### 2.4.3. The EZH2 Expression in Studied U87 and T98G Tumors

Table A6 and Figure 8 show the EZH2 expression data for the U87 and T98G tumor groups.

No significant differences in EZH2 expression were found between the U87-control, U87-3 mM NaDCA, and U87-2 mM VPA–3 mM NaDCA groups. Treatment with 50 µM TMZ significantly reduced the number of EZH2-positive cells in the U87 tumor, and this effect was substantially more significant than that of the combination.

Compared to the T98G-control, the number of EZH2-positive cells was significantly lower in tumors treated with 50 µM TMZ or 2 mM VPA–3 mM NaDCA (no significant difference was found when comparing the latter groups). The treatment with 3 mM NaDCA did not affect EZH2 expression in the T98G tumor.

### 2.5. The Data of SLC12A2, SLC12A5, SLC5A8, CHD1, and CHD2 Expression in Tested U87 and T98G Cell Groups

Expression data for *SLC12A2* (gene encoding Na^+^-K^+^-2Cl^−^ (NKCC1) co-tarnsporter)*, SLC12A5* (gene encoding K^+^-Cl^−^ (KCC2) co-transporter)*,* and *CHD2* (gene encoding the N-cadherin protein) in the U87 and T98G control and treatment groups are shown in Table A7, Table A8 and Table A9 of Appendix A.

#### 2.5.1. *SLC12A2* Expression in U87 and T98G Cells

Table A7 and Figure 9 show the *SLC12A2* expression data for the U87- and T98G-control and treatment cells.

T98G cells expressed significantly higher *SLC12A2* expression than U87 cells. No significant differences in *SLC12A2* expression were found when comparing the U87-control, U87-3 mM NaDCA, U87-50 µM TMZ, and U87-2 mM VPA–3 mM NaDCA groups with the U87-control. Compared to the T98G-control, treatment with 3 mM NaDCA and 2 mM VPA–3 mM NaDCA significantly reduced *SLC12A2* expression, whereas 50 µM TMZ did not affect gene expression.

#### 2.5.2. *SLC12A5* Expression in U87 and T98G Cells

Table A8 and Figure 10 show the *SLC12A5* expression data for the U87- and T98G-control and treated cell groups.

Compared to the U87-control, treatment with 2 mM VPA–3 mM NaDCA significantly increased *SLC12A5* expression in U87 cells. Compared to the U87-control, treatment with 3 mM NaDCA or 50 µM TMZ did not affect the expression of *SLC12A5* in U87 cells. No expression of *SLC12A5* was detected in the T98G-control, T98G-3 mM NaDCA, and T98G-50 µM TMZ groups. Treatment with 2 mM VPA–3 mM NaDCA upregulated *SLC12A5* expression in T98G cells.

#### 2.5.3. *SLC5A8* Expression in U87 and T98G Cells

*SLC5A8* (gene encoding Na^+^-dependent monocarboxylate transporter) expression was not detected in U87- (*n* = 6) and T98G-control cells (*n* = 6 in each group). Treatment with 3 mM NaDCA and 50 µM TMZ did not affect *SLC5A8* expression in U87 cells, while 2 mM VPA–3 mM NaDCA upregulated gene expression in U87 cells (CT of *SLC5A8*—33.39 ± 0.56, CT of *GAPDH*—16.31 ± 0.89; ∆CT 17.09 ± 0.95). Treatment with 3 mM NaDCA, 50 µM TMZ, and 2 mM VPA–3 mM NaDCA did not affect *SLC5A8* expression in T98G cells.

#### 2.5.4. *CDH1* and *CDH2* Expression in U87 and T98G Cells

*CDH1* (gene encoding the E-cadherin protein) was undetectable (silencing) in U87 and T98G cells (*n* = 6 in each group). Treatment with 3 mM NaDCA, 50 µM TMZ, and 2 mM VPA–3 mM NaDCA did not affect *CDH1* expression in U87 cells. Treatment with 3 mM NaDCA and 50 µM TMZ did not affect *CDH1* expression in T98G cells. Treatment with 2 mM VPA–3 mM NaDCA upregulated *CDH1* expression in T98G cells (CT of *CDH1*—34.27 ± 0.43, CT of *GAPDH*—16.80 ± 2.00; ∆CT 17.04 ± 2.49).

Table A9 and Figure 11 show the *CDH2* expression data for the U87- and T98G-control and treated cell groups. The *CDH2* expression of U87 cells was lower than T98G cells. No significant differences in *CDH2* expression were found when comparing the U87-control, U87-3 mM NaDCA, U87-50 µM TMZ, and U87-2 mM VPA–3 mM NaDCA groups. Compared to the T98G-control, treatment with 3 mM NaDCA and 2 mM VPA–3 mM NaDCA significantly reduced *CDH2* expression in T98G cells. T98G cells treated with 2 mM VPA–3 mM NaDCA showed significantly higher *CDH2* expression than those treated with 3 mM NaDCA.

## 3. Discussion

The high mortality in glioblastoma (GBM) patients is related to the tumor’s acquired resistance to chemotherapy with temozolomide (TMZ) [7,50], which results in recurrence and poor prognosis [46], thus highlighting the importance of finding more effective treatments for GBM.

Sodium dichloroacetate (NaDCA) is a specific pyruvate dehydrogenase kinase (PDK) inhibitor that inhibits lactic acid production [51,52] and promotes tumor cell apoptosis [53]. PDKs selectively inhibit the pyruvate dehydrogenase (PDH) complex, which is essential for glycolysis and oxidation mechanisms [13]. PDH inactivation is supported by increased PDK1 expression in GBM cells [54], which is associated with chemotherapy resistance [55].

The anticancer effect of valproic acid (VPA) is associated with inhibition of class I and II histone deacetylases (HDAC) [56]. VPA has been investigated in combination with chemotherapy and radiotherapy [35,57], either at the beginning or after chemoradiotherapy, aiming to increase the efficacy of chemotherapy in GBM patients and improve median overall survival (OS) [58,59]. VPA enhances the efficacy of radiotherapy by increasing the sensitivity of GBM cells [57] and activating the apoptotic response to radiotherapy [60].

The study of U87 and T98G control tumors did not reveal any differences in the frequency of invasion into the chick embryo chorioallantoic membrane (CAM), the expression of neo-angiogenesis in the CAM mesenchyme beneath the tumor, or the expression of proliferating cell nuclear antigen (PCNA) and polycomb inhibitory complex catalytic subunit 2 (EZH2) in the tumor. *SLC5A8* (gene encoding Na^+^-dependent monocarboxylate transporter) and *CDH1* (gene encoding the E-cadherin protein) expression in U87 and T98G control cells was muted (not expressed). T98G cells may be more malignant than U87 cells because *SLC12A2* (gene encoding Na^+^-K^+^-2Cl^−^ (NKCC1) co-tarnsporter) expression was significantly higher in T98G cells than in U87 cells, and *SLC12A5* (gene encoding K^+^-Cl^−^ (KCC2) co-transporter) expression was expressed in U87 cells; it was undetectable in T98G cells, and *CDH2* (gene encoding the N-cadherin protein) expression was significantly higher in T98G control cells than in U87 cells. This study showed that the effect of treatment with NaDCA, valproic acid and NaDCA combination (VPA–NaDCA), or TMZ depended on the test drug and the cell line; this study allowed us to see possible differences or similarities in the effects of the investigated agents on the carcinogenesis of the investigated cell tumors.

The study of U87 tumors on the CAM shows that treatment with 3 mM NaDCA, 2 mM VPA–3 mM NaDCA (hereinafter referred to as VPA–NaDCA), or 50 µM TMZ significantly reduced the incidence of the U87 tumor’s invasion into the CAM and reduced the number of blood vessels in CAM mesenchyme, and the effect of VPA–NaDCA treatment on neo-angiogenesis was significantly greater than that of 3 mM NaDCA and 50 µM TMZ. However, only T98G tumor treatment with 50 µM TMZ significantly reduced the tumor invasion into the CAM. At the same time, neo-angiogenesis was equally blunted by treatment with 50 µM TMZ or VPA–NaDCA of the T98G tumor.

PCNA was initially recognized as a proliferating cell-specific antigen expressed in cell nuclei during the S phase of the cell cycle; later studies and a systematic review show that increased PCNA expression is directly associated with malignancy, poor survival, and advanced GBM stage, and the marker has been proposed as a useful diagnostic or prognostic biomarker or an effective therapeutic target in glioma [61]. Only 50 µM TMZ treatment significantly reduced the number of PCNA-positive cells in the U87 tumor on the CAM. Others reported that TMZ in U87 MG cells at doses (IC_50_ > 200 μM) administered alone exhibited no antiproliferative activity [62]. Regarding the effect of the treatment on PCNA expression, the impact of VPA–NaDCA on the T98G tumor on the CAM is characterized by a synergistic effect of the combination components: the frequency of PCNA-positive cells was lowered considerably in those treated with 50 µM TMZ or VPA–NaDCA tumors, while 3 mM NaDCA did not affect PCNA expression in T98G tumors. Higher doses of 5 mM and 10 mM NaDCA significantly inhibited the PCNA expression in GBM pediatric PBT24 and SF8628 cell tumors on CAM [63].

The unsatisfactory chemotherapeutic effect of TMZ is due to more than 50% of GBM patients being resistant to TMZ [64], mainly due to the high expression of O-6-methylguanine-DNA methyltransferase (MGMT), which alters methylation and preserves tumor cells [65]. Reducing MGMT expression is a key factor in reversing TMZ resistance. The use of high doses of TMZ to reduce MGMT activity still raises the problem of dose-limiting toxicity [65,66]. MGMT regulation has been shown to involve *TP53* gene-encoded p53 protein (p53), which is one of the most comprehensively studied GBM targets [67]. Reduced MGMT expression is associated with p53 activation [64,68]. Compared to the U87 control tumors, only 50 µM TMZ significantly reduced the number of p53-positive cells in the U87 tumor; T98G-50 µM TMZ tumors expressed a significantly decreased p53-positive cell number, while the frequency of p53-positive T98G cells in the tumor did not differ between 2 mM VPA–3 mM NaDCA-treated and control groups.

EZH2 is the enzymatic subunit of polycomb repressive complex 2, a histone N-methyltransferase methylating H3 lysine in mammalian cells, inhibiting transcription [69,70]. EZH2 inhibits genes responsible for suppressing carcinogenesis, and inhibition of EZH2 activity can reduce tumor growth [71]. EZH2 is a target for anticancer drugs [72]; detecting EZH2 expression in tumors using a CAM model is an informative approach [73,74]. Our data showed that treatment with 50 µM TMZ significantly reduced the number of EZH2-positive cells in the U87 tumor, while 3 mM NaDCA or VPA–NaDCA treatment had no effect on EZH2 expression. Otherwise, the number of EZH2-positive cells in tumors was significantly similarly lowered with 50 µM TMZ or VPA–NaDCA treatment, while 3 mM NaDCA did not affect EZH2 expression in the T98G tumor. A meta-analysis of six studies showed that increased EZH2 expression is associated with a poor prognosis in GBM [75]. EZH2 interacts with oncogenes (MYC, PTEN) to influence tumor malignancy by activating the EMT mechanism, initiating GBM cell invasion and metastasis [76].

The Na-K-2Cl co-transporter (NKCC1) carries Na^+^, K^+^, and Cl^−^ into the cell; NKCC1 is encoded by the *SLC12A2* gene [77,78]. The activity of NKCC1 is directly related to GBM cell proliferation [79], which is directly related to the NKCC1 protein expression in human GBM [79,80,81]. Tumor intracellular Cl^−^ concentration is an essential target for anticancer therapy; NKCC1 inhibition can suppress tumor cell proliferation and promote apoptosis [82,83]. The tested IMPs did not change the *SLC12A2* expression in the U87 investigated groups. At the same time, the treatment with 3 mM NaDCA and VPA–NaDCA significantly reduced SLC12A2 expression in T98G cells, whereas 50 µM TMZ did not affect gene expression. Damanskienė et al. found that TMZ significantly increased the expression of *SLC12A2* in the pediatric GBM cells PBT24 and SF8628 in vitro [84]. Activation of *SLC12A2* by TMZ in GBM is dependent on phosphorylation of the WNK kinase protein [85]. Human GBM cells exhibit increased activation of NKCC1 and its two upstream regulatory kinases, WNK1 and OSR1. Knockdown of WNK1 or OSR1 decreases intracellular K^+^ and Cl^−^ levels. TMZ may activate the WNK1/OSR1/NKCC1 signaling pathway and enhance glioma cell migration. Pharmacological inhibition of NKCC1 significantly reduces glioma cell migration after TMZ treatment with drugs that inhibit elements of the WNK1/OSR1/NKCC1 signaling pathway [86]. Inhibition of NKCC1 by bumetanide suppresses glioma cell migration and invasion [87].

A K-Cl co-transporter (KCC2) is identified in central nervous system cells, encoded by *SLC12A5* [88,89]. KCC2 carries K^+^ and Cl^−^ out of the cell [90]. KCC2 expression is reduced in GBM cells; increased KCC2 expression in GBM cells inhibits GBM cell proliferation, i.e., KCC2 is a glioma suppressor [91] and is a prognostic marker for GBM. We showed that VPA–NaDCA treatment increased *SLC12A5* expression in U87 and T98G cells, while 3 mM NaDCA or 50 µM TMZ did not affect *SLC12A5* expression in the cells studied. This indicates the superiority of VPA–NaDCA over TMZ. The effect may be due to the synergistic action of VPA, since VPA upregulated the expression of the gene in U87 but not in T98G cells [92]. *SLC12A5* expression inhibits the proliferation of glioma U251 MG cells, and the gene is a potential prognostic marker for GBM; activation of the co-transporter is an early sign of cell apoptosis [91]. Adult GBM patients with epilepsy have reduced KCC2 expression in histological specimens in the tumor environment [93,94]. Disturbances in the balance between NKCC1 and KCC2 activity may reduce the hyperpolarizing effects of gamma-aminobutyric acid (GABA), which may influence the occurrence of epilepsy in GBM patients. Seizures worsen the course of GBM disease [95]. By activating KCC2 in GBM cells, the effect of drugs may ensure the functioning of GABA A receptors in postsynaptic neurons [89] and could thus help control the intensity of seizures [96].

The monocarboxylate co-transporter (SLC5A8) is an oncosuppressor of human and experimental gliomas, encoded by *SLC5A8*, and its activity depends on the intracellular Na^+^ and Cl^−^ content [97,98]. SLC5A8 activity in male rat duodenal enterocytes depends on NKCC1 activity [99]. SLC5A8 activates cell apoptosis via pyruvate-dependent HDAC suppression [31]. Studies in adult male wild-type c/ebpδ+/- and c/ebpδ-/- mice have shown that SLC5A8 mainly transports short fatty acid chains into the cell [100,101]. The co-transporter carries lactic acid, pyruvate, acetate, propionate, valerate, butyrate, and dichloroacetate (DCA) into the cell [31,97]. We show that *SLC5A8* expression was not detected in U87- and T98G-control cells, and 3 mM NaDCA or 50 µM TMZ did not affect gene expression in these cells. Treatment with 2 mM VPA–3 mM NaDCA upregulated *SLC5A8* expression in U87 cells but not T98G cells. Silencing of *SLC5A8* is associated with DNA methylation [102]; treating tumor cells with DNA demethylating agents increases *SLC5A8* expression [31]. VPA activates genes regulated by DNA methylation [32,103]; VPA increases *SLC5A8* expression in U87 but not T98G cells [92,98].

Cadherins regulate apoptosis, gene expression, cell proliferation, differentiation, and migration [104]. *CDH1* was undetectable (silencing) in U87 and T98G cells. *CDH1* encodes the cell adhesion protein E-cadherin, which is a tumor suppressor [104], and its alterations have been linked to the development of specific epithelial tumors [105]. The N-cadherin gene (*CDH2*) is expressed in GBM and directly correlates with tumor grade [80,104,106]. In gliomas, *CDH1* expression is low or unexpressed due to hypermethylation and decreases with increasing GBM stage [107,108]. This study’s treatment with IMPs did not affect *CDH1* and *CDH2* expression in U87 cells. Treatment with VPA–NaDCA upregulated *CDH1* expression in T98G cells, while 50 µM TMZ did not affect gene expression. The treatment with 3 mM NaDCA and VPA–NaDCA significantly reduced T98G cell’s *CDH2* expression; T98G cells treated with VPA–NaDCA showed significantly lower *CDH2* expression than those treated with TMZ. GBM progression and treatment resistance are linked to the GBM cellular epithelial–mesenchymal transition (EMT) mechanisms, where E-cadherin expression is reduced and N-cadherin expression is increased [109], resulting in the acquisition of a mesenchymal subtype that is more invasive and treatment resistant [104,106].

The limitation of this study is that it only tests cells from two GBM cell lines. Still, data have shown differences in the effects of the investigational medicinal products between the different cell lines. Determining drug efficacy using cell line xenografts on the CAM is limited because commercial GBM cell lines do not accurately represent the genetic and biological heterogeneity of the GBM [109]. It was reported that GBM on the CAM is a model that replicates the histological and genetic features of the patients’ tumor, and the CAM model may enable the prediction of patient clinical outcomes [110]. Therefore, it is essential to research investigational medicines using primary GBM cells. Research shows the advantages of the CAM model. The success rate of GBM on CAM engraftment was 98.3% [110], 38–69% in heterotopic mouse models of patient-derived xenograft gliomas [109,111], and 76–90% in an orthotopic mouse model of a brain tumor [109,112]; however, the problem is that IDH1-mutated transplanted specimens almost entirely fail in mice [111,113,114], and in contrast, patient-derived xenografts on the CAM are practically always successful [110]. Our study suggests that combining the CAM model with the effects of the study drugs on the gene expression of GBM cells would be an approach to evaluating the effectiveness of a drug. The study’s findings underscore the importance of individualized assessment of drug effects and the need for further studies on primary cells from GBM patients to personalize research for effective GBM treatment.

## 4. Materials and Methods

### 4.1. Cell Lines and Culture Conditions

Adult high-grade glioblastoma (GBM) cell lines were investigated: (1) female Caucasian 44-year-old U87 (U-87 MG; ECACC number 89081402), donated by Dr. Arūnas Kazlauskas (Neuro-Oncology and Genetics Laboratory, Institute of Neuroscience, Lithuanian University of Health Sciences, Kaunas, Lithuania), and (2) male Caucasian 61-year-old T98G (T98G; ATTC CRL-1690), donated by Prof. M. M. Alonso (University of Navarra, Pamplona, Spain). The U87 cells were maintained using Dulbecco’s Modified Eagle Medium (DMEM; Gibco, Paisley, UK) supplemented with 10% fetal bovine serum (FBS; Gibco, Paisley, UK), as reported [115]. T98G cells were grown in Advanced Minimum Essential Medium (AMEM; Gibco, Grand Island, NY, USA) supplemented with 2 mM L-glutamine (Glutamax; Gibco, Paisley, UK) and 5% FBS, as described in the product sheet [116]. Media were supplemented with penicillin (1% 100 IU/mL) and streptomycin (100 µg/mL) (P/S; Gibco, Grand Island, NY, USA). Cells were cultured at 37 °C in a humid atmosphere with 5% CO_2_.

### 4.2. U87 and T98G Tumors in the CAM Study Groups

In the U87 and T98G tumor groups, tumor growth and invasion into the chick embryo chorioallantoic membrane (CAM), the number of blood vessels in the CAM, sub-tumoral thickening of the CAM, and the expression of immunohistochemical markers PCNA (proliferating cell nuclear antigen), p53 (*TP53* gene-encoded p53 protein), and EZH2 (polycomb inhibitory complex catalytic subunit 2) were investigated. Cells were treated with 3 mM NaDCA, 50 µM TMZ, or 2 mM VPA–3 mM NaDCA. The group sample sizes are shown in Table 1.

U87 and T98G cells were investigated for *SLC12A2* (gene encoding Na^+^-K^+^-2Cl^−^ (NKCC1) co-tarnsporter), *SLC12A5* (gene encoding K^+^-Cl^−^ (KCC2) co-transporter), *SLC5A8* (gene encoding Na^+^-dependent monocarboxylate transporter), *CDH1* (gene encoding the E-cadherin protein), and *CDH2* (gene encoding the N-cadherin protein) genes in the control group and groups treated with 3 mM NaDCA, 50 µM TMZ, and 2 mM VPA–3 mM NaDCA for 24 h (*n* = 6 per group).

### 4.3. The CAM Model

According to Lithuanian and EU legislation, no approval from the Ethics Committee is required for studies using the CAM model. Fertilized chicken eggs Cobb 500 (Rumšiškės hatchery, Rumšiškės, Lithuania) were incubated for 3 days at 37 °C and 60% air humidity in an egg incubator (Maino Incubators, Oltrona di San Mamette, Como, Italy). Eggs were rolled every hour until day 3 of the embryo development (EDD3). At EDD3, the developing CAM was separated from the eggshell; 2 mL of egg white was aspirated from the blunt end of the egg. A 1 cm^2^ window was drilled in the shell and covered with sterile parafilm. The eggs were re-incubated until GBM cell tumor transplantation on the CAM at EDD7.

U87 or T98G cells were resuspended in 20 µL of type I rat tail collagen (Gibco, New York, NY, USA) (in the control group), and sodium dicholoacetate (NaDCA), combination of valproic acid and NaDCA (VPA–NaDCA), or temozolomide (TMZ) solutions (investigational medicine-treated groups) per 1 × 10^6^ cells. Pieces of an absorbable hemostatic gelatin sponge (Surgispon, Aegis Lifesciences, Gujarat, India) having a size of 9 mm^3^ (3 × 3 × 1 mm) were formed using a blade. A quantity of 20 µL of cell suspension mixture was pipetted onto each piece of sponge. As stated by Ribatti, the gelatin sponge is suitable for the delivery of tumor cell suspensions onto a CAM surface [117]. At the EDD7, the formed tumors were transplanted on the CAM near the largest blood vessels. The tumor’s structural changes were monitored in vivo using a stereomicroscope (SZX2-RFA16, Olympus, Tokyo, Japan) during the period of EDD9–12. Images of the tumors were captured with a digital camera (DP 92, Olympus, Tokyo, Japan) and CellSens Dimension 1.9 Imaging Software. At EDD12, the CAM blood vessel was injected with 0.2 mL of fluorescein isothiocyanate-dextran (Sigma-Aldrich, St. Louis, MO, USA) at 5 mg/mL. The tumors with the CAM were harvested and photographed in daylight and with a green fluorescent protein (GFP) filter exposed to UV light using the software.

### 4.4. Investigational Medicinal Products

The investigational medicinal preparations used for the study were sodium dichloroacetate (NaDCA; Sigma-Aldrich, Steinheim, Germany), sodium valproate (VPA; Sigma-Aldrich, Steinheim, Germany), and temozolomide (TMZ; Sigma-Aldrich, St. Louis, MO, USA). The investigational medicinal product combines NaDCA and VPA. The combination of these products is our filed patent that covers VPA–NaDCA as a novel medicinal product for treating cancer (Official bulletin of the State Patent Bureau of the Republic of Lithuania, number 6874, filed on 17 April 2020) [118]; a European patent application was submitted (European patent application number 21168796.7, filed on 16 April 2021) [119]. The doses of NaDCA or VPA were chosen as previously reported, using investigational drugs for monotherapy [73,115,120]. A 50 µM TMZ dose correlates with the mean TMZ plasma concentration in TMZ-treated patients [121].

### 4.5. Histological and Immunohistochemical Examination of the Tumor

At EDD12, the tumors with the CAM were harvested, fixed in 10% neutral-buffered formalin for 24 h, and embedded in paraffin. The Obtained 3 µm sections were stained using the hematoxylin and eosin (H-E) method (Sigma-Aldrich, Darmstadt, Germany) and immunohistochemically (IHC) for PCNA, p53, and EZH2 markers. Sections were mounted on glass slides coated with poly-L-lysine (Thermo Fisher Scientific, Branchburg, NJ, USA). Deparaffinization of sections in xylene and rehydration in progressive alcohol concentrations were followed by heat-induced antigen retrieval with EnVision FLEX Target Retrieval Solution (K8002, Dako, Glostrup, Denmark) (95 °C, 20 min, pH 9). The Shandon Coverplate Technology (Thermo Fisher Scientific, Branchburg, NJ, USA) was used for the staining. Endogenous peroxidase blocking was performed using EnVision FLEX Peroxidase Blocking Reagent (SM 801, Dako, Glostrup, Denmark) for 10 min. The preparations were incubated with primary antibodies (dilution 1:100) for 30 min at room temperature. Primary antibodies to PCNA (PC10, Thermo Fisher Scientific, Branchburg, NJ, USA), p53 (aa 211–220, clone 240, CBL 404, Merck, Darmstadt, Germany), and KMT6/EZH2 (phospho S21, ab 84989, Abcam, Cambridge, UK) were applied to detect PCNA-, p53-, and EZH2-positive cells in the tumor tissue. Each array of slides was used as the positive control; the primary antibody was omitted to achieve a negative control. The IHC reaction was revealed using polymer dextran labeled with horseradish peroxidase and conjugated to a secondary antibody (mouse) and a linker (SM 802 and SM 804, respectively; Dako, Glostrup, Denmark) for 30 min at room temperature. Positive reactions were visualized using 3,3′-diaminobenzidine-containing chromogen (DAB, DM827, Dako, Glostrup, Denmark). After each stage, Tween 20 containing tris-buffered saline (DM 831, Dako, Glostrup, Denmark) was used as a wash buffer. Slides were counterstained with hematoxylin (Sigma-Aldrich, Taufkirchen, Germany).

H-E-stained and IHC-stained slides were visualized and captured by a light microscope (BX 40F4, Olympus, Tokyo, Japan) and a digital camera (XC 30, Olympus, Tokyo, Japan) with Cell Sens Dimension 1.9 software.

Tumors stained with H-E were divided into invasive and non-invasive. The tumor invasion into the CAM was classified as destroying the chorionic epithelium (ChE) or/and the migration of tumor cells into the mesenchyme of the CAM. The non-invasive tumor was located on top of the CAM, and the integrity of the CAM chorionic epithelium was intact. The tumor invasion was studied using H-E slides at 10× and 40× magnifications.

The CAM thickening and the number of vessels were assessed by capturing the CAM stained with H-E at 4× magnification in the location under the tumor. The thickness of the CAM was measured in ten areas (µm), and the mean thickness of the CAM was calculated; blood vessels with a larger diameter than 10 µm were counted. All calculations were performed in the same length of CAM (1792 µm).

To assess the PCNA, p53, and EZH2 expression in the tumor, 2 vision fields were selected randomly (plot area—23,863.74 µm^2^) in the IHC-stained tumor and were captured at 40× magnification. All cells and cells positive with the studied marker were counted, and the percentage (%) of marker-positive cells in each tumor was calculated.

### 4.6. Gene Expression Analysis

U87 and T98G cells were treated with 3 mM NaDCA, 50 µM TMZ, and 2 mM VPA–3 mM NaDCA for 24 h. Control cell groups were cultivated in cell culture medium according to the cell line. A TRIzol Plus RNA Purification Kit (Life Technologies, Carlsbad, CA, USA) was used for total RNA extraction. Purity of total RNA and concentration were measured with a NanoDrop 2000 spectrophotometer (Thermo Scientific, Branchburg, NJ, USA). The integrity of total RNA (RIN) was evaluated with an Agilent 2100 Bioanalyzer (Agilent Technologies, Santa Clara, CA, USA) using the Agilent RNA 6000 Nano Kit (Agilent Technologies, Santa Clara, CA, USA). RNA expression was investigated for genes *SLC12A2* (Hs00169032_m1; 97 bp), *SLC12A5* (Hs00221168_m1; 80 bp), *SLC5A8* (Hs00377618_m1; 88 bp), *CDH1* (Hs01023894_m1; 61 bp), and *CDH2* (Hs00983056_m1; 66 bp). The glyceraldehyde-3-phosphate dehydrogenase (*GAPDH*) gene was used (Hs02786624_g1; 157 bp) as a reference gene. The CT cut-off value was set at 35. Following the manufacturer’s instructions, total RNA reverse transcription was performed using 100 ng of RNA in a 20 µL reaction volume on a Biometra T Advanced thermal cycler (Analytik Jena AG, Jena, Germany) with the High-Capacity cDNA Reverse Transcription Kit and RNase Inhibitor (Applied Biosystems, Waltham, MA, USA). Real-time polymerase chain reaction (PCR) was performed on an Applied Biosystems 7900 Fast Real-Time PCR System (Applied BioSystems, Waltham, MA, USA) with TaqMan assays (Applied Biosystems, Pleasanton, CA, USA) according to the manufacturer’s instructions. The reactions were run in triplicate with 4 μL of cDNA template in a reaction volume of 20 μL (TaqMan Universal Master Mix II—10 μL, no UNG (Applied Biosystems, Vilnius, Lithuania), TaqMan Gene Expression Assay 20x—1 μL (Applied Biosystems, Pleasanton, CA, USA), and nuclease-free water—5 μL (Invitrogen, Paisley, UK); the program was set at 95 °C for 10 min, then 45 cycles of 95 °C for 15 s, followed by 60 °C for 1 min.

### 4.7. Statistical Analysis

Statistical analysis was conducted and graphs generated using Graph Pad Prism 9 software (Graph Pad Software, Inc., San Diego, CA, USA). The CT threshold cycle value was normalized to the control GAPDH for gene expression analysis, and the ΔCT value was calculated accordingly. The Livak (2^−ΔΔCT^) method [122] was used to calculate the expression ratio between the control and treated groups of the tested genes. The frequency of tumor invasion into the CAM is expressed as a percentage (%), and the chi-square test was used to compare the frequency of tumor invasion into the CAM between the tested groups. The Shapiro–Wilk test was used to verify the assumption of normality. Data on immunohistochemical markers, the number of vessels in the CAM, and sub-tumoral thickening of the CAM are presented as median and range (values of minimum and maximum). The difference between the two groups was evaluated using the nonparametric Mann–Whitney U test. Statistical significance was defined as *p* < 0.05.

## 5. Conclusions

This study showed similarities and differences in the effects of NaDCA, VPA–NaDCA, and TMZ on growth, neo-angiogenesis, and PCNA, EZH2, and p53 expression in the tumor tissue of U87 and T98G tumors on the CAM, and that these effects were cell line dependent. The combined VPA and NaDCA components act synergistically. The effects of VPA–NaDCA and TMZ on the expression of *SLC12A2*, *SLC12A5*, *SLC5A8*, *CDH1*, and *CDH2* in U87 and T98G cells were cell line dependent, and the efficacy data indicate a superior effect of VPA–NaDCA compared to TMZ.

## 6. Patents

The combination of VPA and NaDCA products is for the treatment of cancer (Official Bulletin of the State Patent Bureau of the Republic of Lithuania, No. 6874, filing date: 17 April 2020). A European patent application has been submitted (European Patent Application No. 21168796.7, filing date: 16 April 2021).

## Figures and Tables

**Figure 1 ijms-26-06784-f001:**
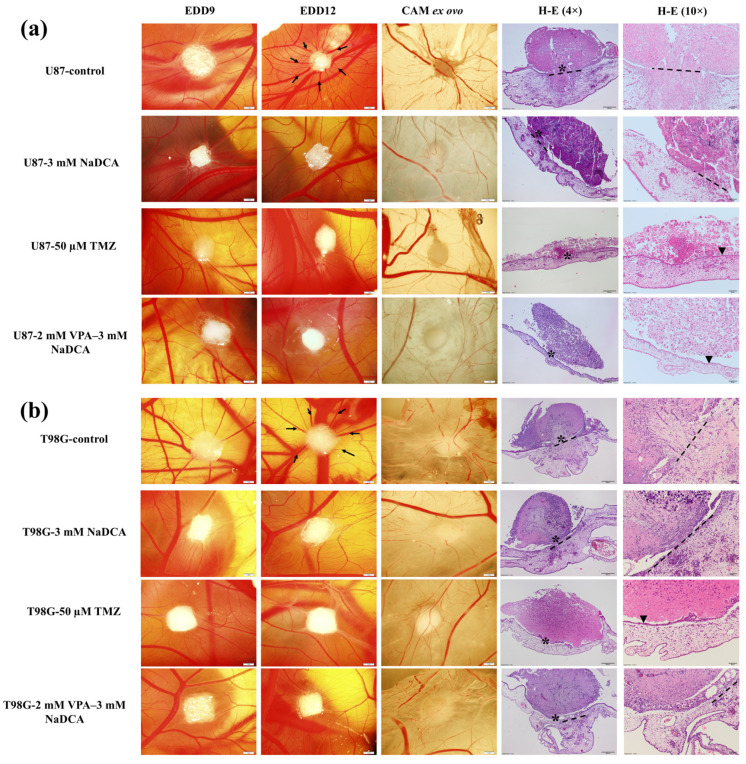
Stereomicroscopic U87 (**a**) and T98G (**b**) tumors on the CAM, the ex ovo tumor with the CAM, and the tumor histologic images stained with H-E at 4× and 10× magnifications. Arrows indicate the “spoked wheel”; dotted line—disruption of chorionic epithelium; asterisk—the enlarged region in H-E 10× pictures; arrowhead—intact chorionic epithelium. CAM—chick embryo chorioallantoic membrane; EDD—day of embryo development; H-E—hematoxylin and eosin. EDD9, EDD12, CAM ex ovo scale bar is 1 mm; H-E 4× preparation scale bar is 200 µm, 10× magnification—50 µm.

**Figure 2 ijms-26-06784-f002:**
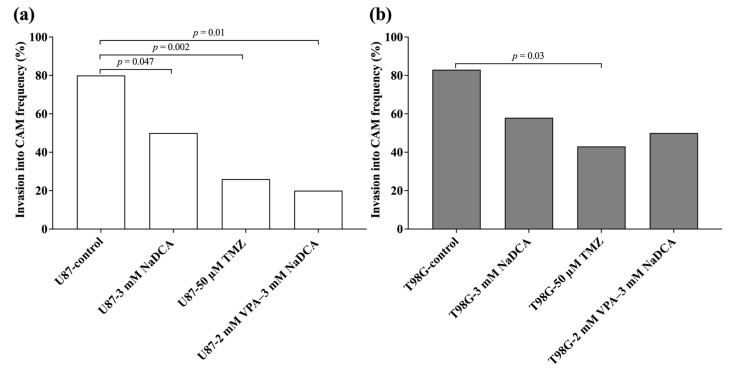
Invasion rate of U87 (**a**) and T98G (**b**) tumors into the CAM.

**Figure 3 ijms-26-06784-f003:**
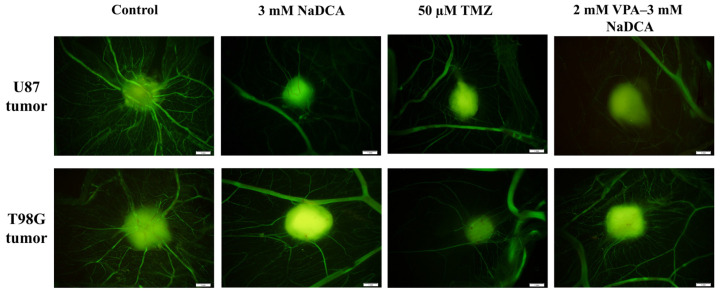
Fluorescent stereomicroscopic images of U87 and T98G tumors. The scale bar—1 mm.

**Figure 4 ijms-26-06784-f004:**
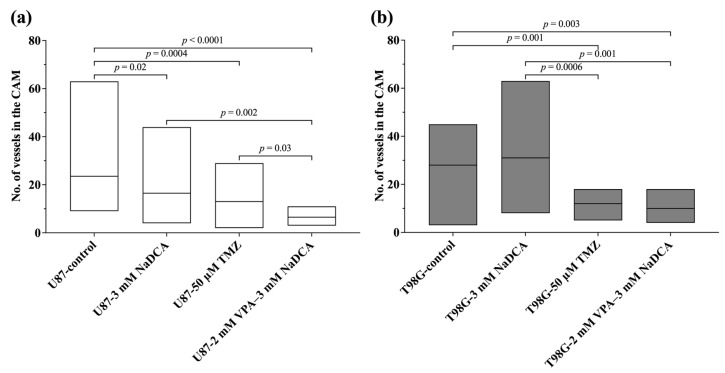
The number of blood vessels in the CAM under the U87 (**a**) and T98G (**b**) study tumors.

**Figure 5 ijms-26-06784-f005:**
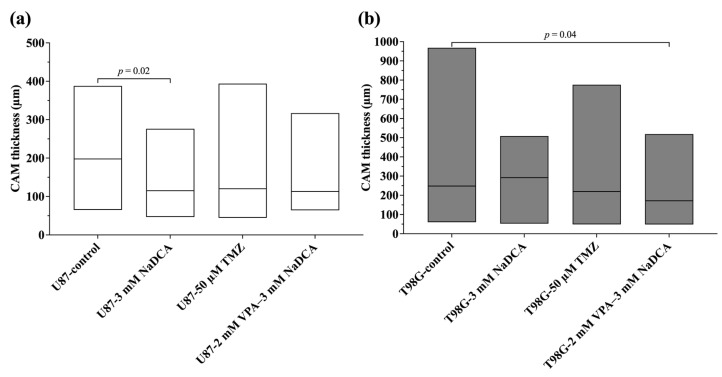
The CAM thickness under the U87 (**a**) and T98G (**b**) control and treated groups.

**Figure 6 ijms-26-06784-f006:**
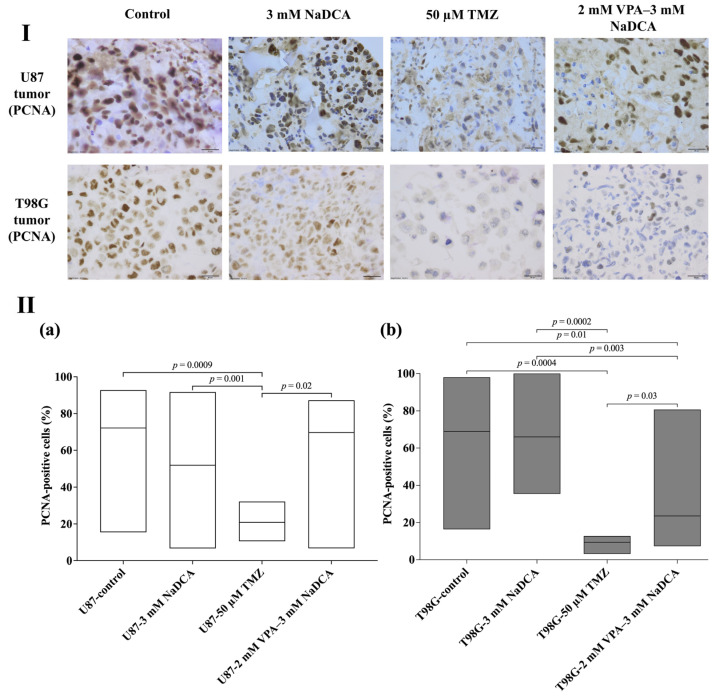
The PCNA expression in the groups of U87 and T98G control and treated tumors. (**I**) Dark brown nuclei indicate a PCNA-positive cell. Scale bar—20 µm. (**II**) The percentage of PCNA-positive cells in U87 (**a**) and T98G (**b**) tumors. PCNA—proliferating cell nuclear antigen.

**Figure 7 ijms-26-06784-f007:**
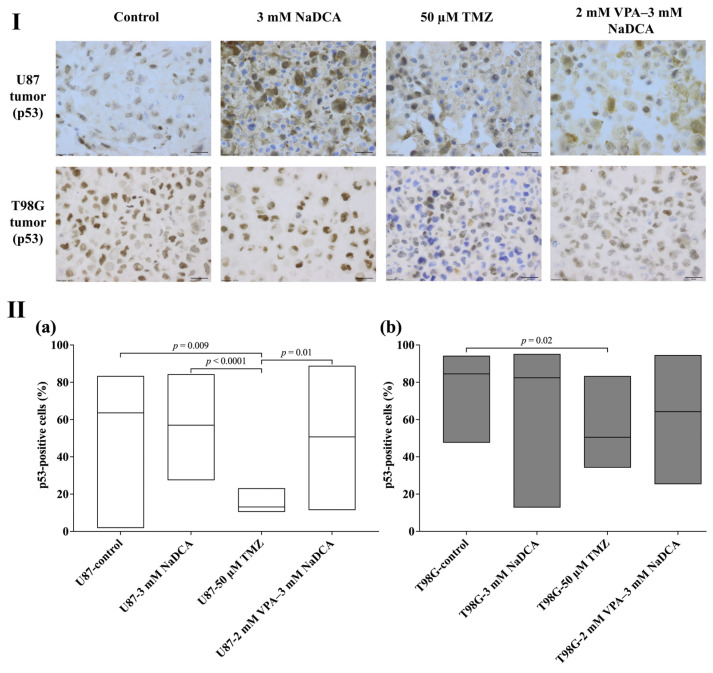
The p53 expression in the U87 and T98G control and treated tumors. (**I**) Dark brown nuclei indicate a p53-positive cell. Scale bar—20 µm. (**II**) The percentage of p53-positive cells in U87 (**a**) and T98G (**b**) tumors. p53—*TP53* gene-encoded p53 protein.

**Figure 8 ijms-26-06784-f008:**
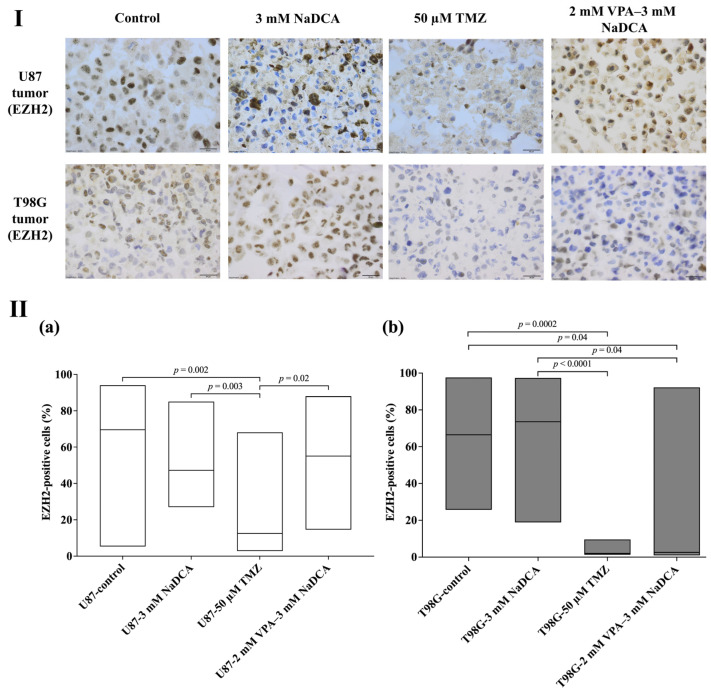
The EZH2 expression in the U87 and T98G control and treated tumors. (**I**) Dark brown nuclei indicate an EZH2-positive cell. Scale bar—20 µm. (**II**) The percentage of EZH2-positive cells in U87 (**a**) and T98G (**b**) tumors. EZH2—polycomb inhibitory complex catalytic subunit 2.

**Figure 9 ijms-26-06784-f009:**
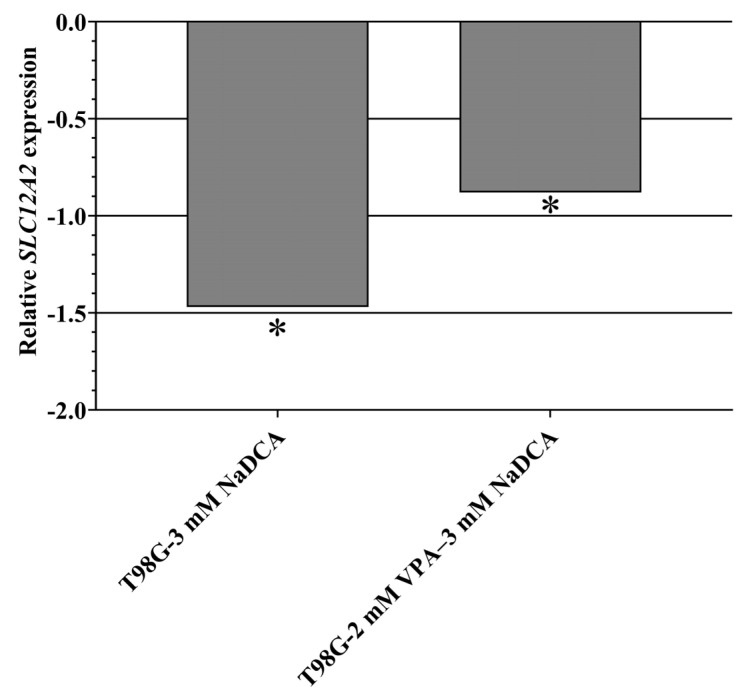
The *SLC12A2* expression in the T98G cells tested: relative *SLC12A2* expression (Log_2_(2^−ΔΔCT^)) of the T98G cell groups tested. *SLC12A2*—gene encoding Na^+^-K^+^-2Cl^−^ (NKCC1) co-tarnsporter. * *p* < 0.05, compared to the control.

**Figure 10 ijms-26-06784-f010:**
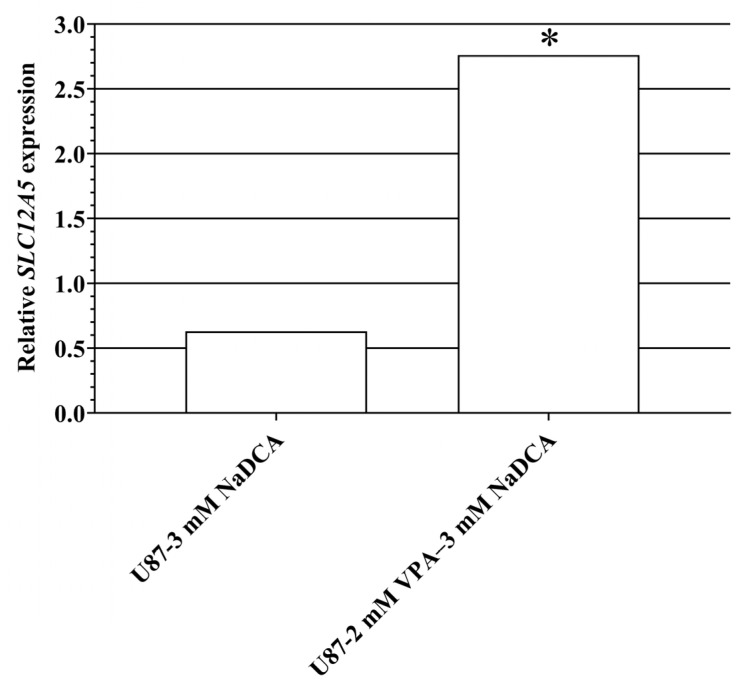
The *SLC12A5* expression in the U87 cells tested: relative *SLC12A5* expression (Log_2_(2^−ΔΔCT^)) of the U87 cell groups. *SLC12A5*—gene encoding K+-Cl– (KCC2) co-transporter. * *p* < 0.05, compared to the control.

**Figure 11 ijms-26-06784-f011:**
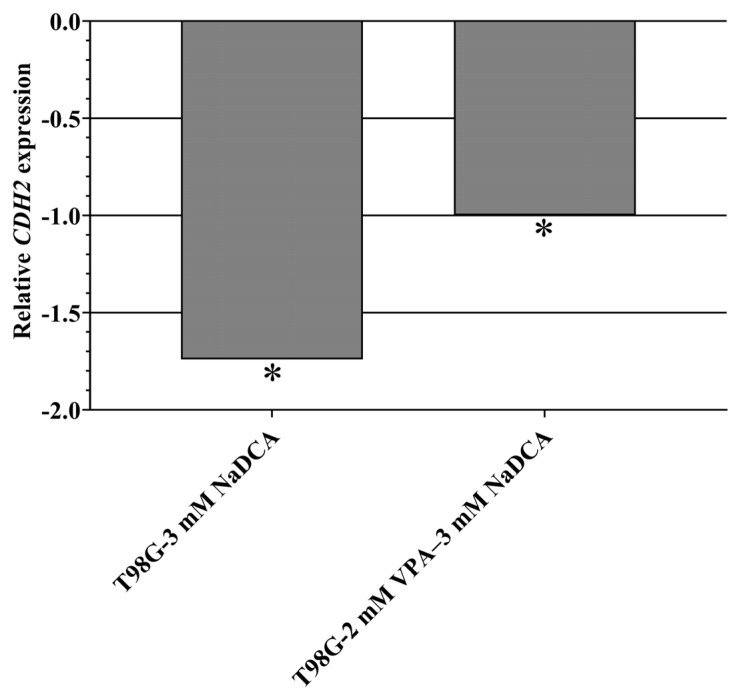
The *CDH2* expression in the T98G cells tested: relative *CDH2* expression (Log_2_(2^−ΔΔCT^)) of the tested T98G cells. *CDH2*—gene encoding the N-cadherin protein. * *p* < 0.05, compared to the control.

**Table 1 ijms-26-06784-t001:** Sample size of U87 and T98G tumors on the CAM and tumor IHC marker expression in the groups studied.

Control and Treated Study Group	Invasion, No. of Vessels, CAM Thickness		PCNA		p53		EZH2	
	*n*							
	U87	T98G	U87	T98G	U87	T98G	U87	T98G
Control	20	12	13	9	15	9	17	9
3 mM NaDCA	20	12	18	10	20	11	19	11
50 µM TMZ	15	16	8	6	7	7	8	7
2 mM VPA–3 mM NaDCA	10	12	10	8	10	8	10	9

## Data Availability

Data will be made available upon request.

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
