# Peer review of "In Vivo and In Vitro Experimental Study Comparing the Effect of a Combination of Sodium Dichloroacetate and Valproic Acid with That of Temozolomide on Adult Glioblastoma"

_ijms, 2025, doi:10.3390/ijms26146784_

Round 1
Reviewer 1 Report
Comments and Suggestions for Authors
Type of manuscript: Article
In vivo and in vitro experimental study comparing the effect of a combination of sodium dichloroacetate and valproic acid with that of temozolomide on adult glioblastoma
The manuscript presents a valuable study for similarities and differences in the effects of NaDCA, VPA–NaDCA and TMZ on growth, neoangiogenesis, PCNA, EZH2 and p53 expression in tumor tissue of U87 and T98G tumors on CAM, and that these effects were cell line dependent.
The combined VPA and NaDCA components act synergistically. The effects of VPA–NaDCA and TMZ on the expression of SLC12A2, SLC12A5, SLC5A8, CDH1, and CDH2 in U87 and T98G cells were cell line dependent, and the efficacy data indicate a superior effect of VPA–NaDCA compared to TMZ.
The interpretation of results and study conclusions supported by the data.
Comments for the author:
I suggest that a valuable study for similarities and differences in the effects of NaDCA, VPA–NaDCA , TMZ with NaDCA and TMZ with VPA on the U87 and T98G cell tumors on CAM,
The interpretation of results uses tables and curves. Please remove figures 2, 4, 5,6,7,8 use tables only.
Author Response
Reviewer‘s comments and authors' responses and made corrections to comments
We thank the Reviewers for their comments, notes and questions. The text of the manuscript has been revised in line with comments. The corrections have been made with Track Changer and are visible. The English text has been revised and corrected. Below are our disclosures, along with the corrections made to the manuscript text in response to the comments of all Reviewers.
Review Report Form REVIEWER 1
Open Review
Comments and Suggestions for Authors
Type of manuscript: Article
In vivo and in vitro experimental study comparing the effect of a combination of sodium dichloroacetate and valproic acid with that of temozolomide on adult glioblastoma
The manuscript presents a valuable study for similarities and differences in the effects of NaDCA, VPA–NaDCA and TMZ on growth, neoangiogenesis, PCNA, EZH2 and p53 expression in tumor tissue of U87 and T98G tumors on CAM, and that these effects were cell line dependent.
The combined VPA and NaDCA components act synergistically. The effects of VPA–NaDCA and TMZ on the expression of SLC12A2, SLC12A5, SLC5A8, CDH1, and CDH2 in U87 and T98G cells were cell line dependent, and the efficacy data indicate a superior effect of VPA–NaDCA compared to TMZ.
The interpretation of results and study conclusions is supported by the data.
Comments for the author:
I suggest that a valuable study for similarities and differences in the effects of NaDCA, VPA–NaDCA , TMZ with NaDCA and TMZ with VPA on the U87 and T98G cell tumors on CAM,
The interpretation of results uses tables and curves. Please remove figures 2, 4, 5,6,7,8 use tables only.
Answer
Thank you for the important comment. We agree that Figures 2, 4, 5, 6, 7 and 8 below the respective Tables 1, 2, 3, 4, 5, 6 with the corresponding data are a repetition of the data. We hope we have resolved this issue by relocating the Tables to the Appendix and modifying the table numbering as A1–A6 in Appendix A of the manuscript (pages 27−29). In the text of the Results section, we have inserted a sentence indicating that the data for the PCNA, p53 and EZH2 markers tested in the tumor are in the corresponding Tables A4−6 in Appendix A (L197–L198).
We are grateful to the reviewer for his comments and for the opportunity to correct the article.
Sincerely,
Donatas Stakišaitis

Reviewer 2 Report
Comments and Suggestions for Authors
The aim of this study is to evaluate and compare, through both in vivo (CAM model) and in vitro (GBM U87 and T98G 99 glioblastoma cell lines) experimental models, the therapeutic effects of a combination of sodium dichloroacetate and valproic acid with those of the standard chemotherapeutic agent temozolomide in the treatment of adult glioblastoma.
After a thorough review of your study, I would like to address a few questions, starting with the Materials and Methods section.
In line 551, please provide more details on how you treated the cells with 3 mM NaDCA, 50 µM TMZ, and 2 mM VPA–3 mM NaDC.. How many cells were used, how were the compounds reconstituted, and how were these doses selected?
Line 483 What criteria were used to establish the concentrations of NaDCA, VPA, and TMZ?
Line 486 Was the sponge employed as a cellular scaffold in this study? Were any potential interactions examined as well, or what was its specific intended purpose?I would appreciate it if you could provide more specific details.
Line 482-485 - Please reconsider this paragraph, as it is difficult to understand in its current form and appears to lack a clear logical structure.
Could you please specify the cell concentration mentioned in line 487?
Could you please clarify what is meant by 'formed tumor' in line 488? Could you please clarify what is meant by 'were grafted'?
Please use the correct formatting for in vivo and in vitro.
Could you please elaborate on the procedure used for fluorescein injection as described in line 492?
Could you please clarify what is meant by 'photographed in daylight'?
I kindly request a thorough reevaluation of lines 446-505, as the narrative flow is confusing and lacks coherence in its present form.
I kindly suggest that you revise the descriptions of the images (fig.6,7,8), graphs, and tables to ensure clarity and completeness, and that you enhance the graphical quality where needed.
Although the Results and Discussion sections are generally well presented, the manuscript lacks a clear and cohesive narrative that would guide the reader through the experimental progression. It is recommended that the authors revise and restructure these sections to enhance clarity, with particular emphasis on the novel contributions of the study and its limitations. Furthermore, the manuscript would benefit from a more detailed justification of the use of the CAM model in comparison to alternative experimental models. The Abstract should also be revised to better reflect these improvements and provide a concise summary aligned with the study's key findings and significance.
Author Response
Reviewer‘s comments and authors' responses and made corrections to comments
We thank the Reviewers for their comments, notes and questions. The text of the manuscript has been revised in line with comments. The corrections have been made with Track Changer and are visible. The English text has been revised and corrected. Below are our disclosures, along with the corrections made to the manuscript text in response to the comments of all Reviewers.
Review Report Form REVIEWER 2
Open Review
Comments and Suggestions for Authors
The aim of this study is to evaluate and compare, through both in vivo (CAM model) and in vitro (GBM U87 and T98G 99 glioblastoma cell lines) experimental models, the therapeutic effects of a combination of sodium dichloroacetate and valproic acid with those of the standard chemotherapeutic agent temozolomide in the treatment of adult glioblastoma.
After a thorough review of your study, I would like to address a few questions, starting with the Materials and Methods section.
In line 551, please provide more details on how you treated the cells with 3 mM NaDCA, 50 µM TMZ, and 2 mM VPA–3 mM NaDCA. How many cells were used, how were the compounds reconstituted, and how were these doses selected?
Answer
The doses of the IMPs were chosen in line with our previous studies where NaDCA or VPA were used as monotherapy. This is a continuation of our studies. In general, our policy is that if the dosage of NaDCA in the solution for monotherapy in vitro studies exceeds 10 mM, the product is considered ineffective. It is worth noting that several publications report NaDCA concentrations exceeding 10 mM, 15 mM, or even more in in vitro studies. However, these studies are irrelevant because the concentration of Na ions in solution corresponds to hypernatremia (when added to the concentration of Na ions in the medium). For the study NaDCA monotherapy and combination therapy, the same 3 mM NaDCA concentration was chosen to assess synergism.
The text of the article has been supplemented with sentences and additional references: „The doses of the investigational drugs were chosen as previously reported, using NaDCA or VPA for monotherapy [73, 115, 121]. 50 µM TMZ dose was selected because it correlates with the mean TMZ plasma concentration in TMZ-treated patients [122]“ (L537–L540).
For gene expression analysis, 0.5×106 U87 cells and 0.7×106 T98G cells were seeded, as indicated in Materials and Methods. After 24 hours of incubation, the media were replaced with a new medium containing 3 mM NaDCA, 50 µM TMZ, or VPA–NaDCA. The cells were incubated with media containing the investigative medicinal product, i.e., treated for 24 hours.
For stock solutions, NaDCA and VPA–NaDCA were dissolved in cell culture medium, and TMZ was dissolved in DMSO; further, working IMP solutions were prepared in culture medium. It is important to notice that the final DMSO concentration was less than 0.1 %, so, according to literature (Herbener VJ, Burster T, Goreth A, Pruss M, von Bandemer H, Baisch T, Fitzel R, Siegelin MD, Karpel-Massler G, Debatin KM, Westhoff MA, Strobel H. Considering the Experimental use of Temozolomide in Glioblastoma Research. Biomedicines. 2020 Jun 4;8(6):151. doi: 10.3390/biomedicines8060151. PMID: 32512726; PMCID: PMC7344626.), it is safe for working with cell cultures.
Line 483 What criteria were used to establish the concentrations of NaDCA, VPA, and TMZ?
Answer
The criteria to establish the concentrations of NaDCA, VPA and TMZ were chosen as previously reported by us, please see the revised Manuscript (L537–L540).
Line 486 Was the sponge employed as a cellular scaffold in this study? Were any potential interactions examined as well, or what was its specific intended purpose?I would appreciate it if you could provide more specific details.
Answer
No potential sponge interactions were examined as the sponge was used as a scaffold in the formation of every single tumor. It was applied as a framework to prevent the cells from scattering when transplanted onto the CAM.
The text of the article has been supplemented with a sentence and additional reference: „As stated by Ribatti, the gelatin sponge is suitable for the delivery of tumor cell suspensions onto a CAM surface [117]“ (L517-L518).
Line 482-485 - Please reconsider this paragraph, as it is difficult to understand in its current form and appears to lack a clear logical structure.
Thank you for your notice. We have made changes in the mentioned paragraph.
(L511–L514).
Could you please specify the cell concentration mentioned in line 487?
1×106 cells as shown in Manuscript (L514).
Could you please clarify what is meant by 'formed tumor' in line 488? Could you please clarify what is meant by 'were grafted'?
Answer
The formed tumor consists of a 20 μL mixture of cells, rat tail collagen type I (also, in the treatment group - an investigative medicinal preparation diluted in cell culture media or DMSO), pipetted onto the absorbable hemostatic gelatin sponge. By grafting, we meant that the tumor was transplanted onto the CAM’s surface. We have replaced the word „grafted“with the word „transplanted“ in the Manuscript (L519).
Please use the correct formatting for in vivo and in vitro.
Answer
Thank you for the remark. We have corrected the formatting.
Could you please elaborate on the procedure used for fluorescein injection as described in line 492?
Answer
Fluorescein isothiocyanate-dextran was injected using an insulin syringe into the major CAM blood vessel. We have enhanced the graphical quality in Figure 3 (page 7).
Could you please clarify what is meant by 'photographed in daylight'?
Answer
The CAM was photographed under routine microscope lighting, and no filters were used.
I kindly request a thorough reevaluation of lines 446-505, as the narrative flow is confusing and lacks coherence in its present form.
Answer
Thank you for your observation, we have made changes in the mentioned lines.
(L474-540 in the fixed version).
I kindly suggest that you revise the descriptions of the images (fig.6,7,8), graphs, and tables to ensure clarity and completeness, and that you enhance the graphical quality where needed.
Answer
Thank you for your comment. We have corrected Figures 6, 7, and 8, as well as the graphs. We have enhanced the graphical quality in Figures 1, 3, 6, 7, 8. To ensure clarity and completeness, we have relocated all tables (A1-A9) to Appendix A.
Although the Results and Discussion sections are generally well presented, the manuscript lacks a clear and cohesive narrative that would guide the reader through the experimental progression. It is recommended that the authors revise and restructure these sections to enhance clarity, with particular emphasis on the novel contributions of the study and its limitations. Furthermore, the manuscript would benefit from a more detailed justification of the use of the CAM model in comparison to alternative experimental models.
Answer
Thank you for your valuable comment. We have added text to the discussion (L455-L472).
The Abstract should also be revised to better reflect these improvements and provide a concise summary aligned with the study's key findings and significance.
Answer
The journal allows a limited number of characters in the abstract. The study encompasses a substantial amount of data, which we present briefly to highlight the differences between the cells under study. As the other reviewers had no comments on the abstract, we hope to avoid making changes.
We are grateful to the Reviewer for their critical comments, on which we were able to improve the article.
Sincerely,
Donatas Stakišaitis

Reviewer 3 Report
Comments and Suggestions for Authors
In this study, authors compared the effect of the single drugs: temozolomide (TMZ) and sodium dichloroacetate (NaDCA) to the combination therapy: sodium dichloroacetate + valproic acid (VPA). The authors used a novel patient-derived xenograft (PDX) model system - chick embryo chorioallantoic membrane (CAM) and tested a number of parameters as: histology, vascular network, and invasion frequency, as well as expression of PCNA, EZH2, p53, SLC12A2, SLC12A5, SLC5A8, CHD1, and CHD2. The authors tested two patient-derived high-grade glioblastoma cell lines: U87 and T98G. The general conclusion is that combination therapy often works better than single-agent therapy, but the effect varies greatly between the cell lines.
This research sheds light on the important and pressing issue of the diversity in high-grade glioma cancers and drug resistance. It's novel and of interest to oncologists and especially neuro-oncologists.
I'd recommend publishing it after minor corrections.
Corrections and suggestions:
Substantial:
1) NaDCA is used to increase the sensitivity of GBNs to TMZ, and VPA is used to treat NaDCA-related neuropathy, so why did the authors not test triple-therapy: TMZ + NaDCA + VPA?
2) Throughout the whole study, the authors have a pretty big difference in the sample size (10-12 to 20 samples). The p-value will change when you increase the number of samples, so it's questionable if the authors can use a universal p<0.05 as a significance criterion. Please consult the statistician and either downsample to the same sample size or use another appropriate analysis. Please add appropriate clarification in the Materials and Methods.
Minor:
3) Figure 1. Stereomicroscopic images: please add arrows or some other accents to show (i) a "spoked wheel" and (ii) the intact or disrupted integrity of the chorionic epithelium on the images.
4) Figure 1. H&E images: please ass extra column with the enlarged H&E images (and please mark the region that is enlarged); it's hard to see anything on the current images.
5) Table 1: p<0.05 is an arbitrary number that was agreed to be a significance criterion. ap = 0.047 is technically passing as a significant difference. What was the p-value for the T98G-2 mM VPA-3 mM NaDCA?
6) Figure 2: Please add either the data points to the bar graph, or ideally, present the data as a box plot.
7) Figure 6I: The red arrows are confusing. I'd suggest removing the arrows from "zoom-out" panels and adding the enlarged "zoom-in" of the significant regions showing the stain-positive and stain-negative cells on the enlarged images.
8) Figure 7I: The same as Fig. 6I, please remove the red arrows from "zoom out" panels and add enlarged panels, showing the stain-positive and negative cells there.
9) Line 451: "Kaunas" is missing in the "Lithuanian University of Health Sciences, Lithuania"
Author Response
Reviewer‘s comments and authors' responses and made corrections to comments
We thank the Reviewers for their comments, notes and questions. The text of the manuscript has been revised in line with comments. The corrections have been made with Track Changer and are visible. The English text has been revised and corrected. Below are our disclosures, along with the corrections made to the manuscript text in response to the comments of all Reviewers.
Review Report Form REVIEWER 3
Open Review
Comments and Suggestions for Authors
In this study, authors compared the effect of the single drugs: temozolomide (TMZ) and sodium dichloroacetate (NaDCA) to the combination therapy: sodium dichloroacetate + valproic acid (VPA). The authors used a novel patient-derived xenograft (PDX) model system - chick embryo chorioallantoic membrane (CAM) and tested a number of parameters as: histology, vascular network, and invasion frequency, as well as expression of PCNA, EZH2, p53, SLC12A2, SLC12A5, SLC5A8, CHD1, and CHD2. The authors tested two patient-derived high-grade glioblastoma cell lines: U87 and T98G. The general conclusion is that combination therapy often works better than single-agent therapy, but the effect varies greatly between the cell lines.
This research sheds light on the important and pressing issue of the diversity in high-grade glioma cancers and drug resistance. It's novel and of interest to oncologists and especially neuro-oncologists.
I'd recommend publishing it after minor corrections.
Corrections and suggestions:
Substantial:
1) NaDCA is used to increase the sensitivity of GBNs to TMZ, and VPA is used to treat NaDCA-related neuropathy, so why did the authors not test triple-therapy: TMZ + NaDCA + VPA?
Answer
Thank you for the suggestion. This work aimed to compare the effect of the investigational drugs, including the combination, with that of temozolomide. Clinical trials of NaDCA in the treatment of GBM are currently underway (https://clinicaltrials.gov/study/NCT05120284?cond=Glioblastoma&term=dichloroacetate&rank=1). We aim to compare the newly developed combination with the effects of other IMPs. For TMZ+NaDCA+VPA combination studies, it is essential to note that TMZ can activate the WNK1/OSR1/NKCC1 pathway, which exacerbates the course of GBM. It is worth noting that VPA may also increase NKCC1 expression in certain patients. Therefore, further studies are needed to address the individualised feasibility of this combination. Such research requires a separate study. Thank you for your suggestion.
2) Throughout the whole study, the authors have a pretty big difference in the sample size (10-12 to 20 samples). The p-value will change when you increase the number of samples, so it's questionable if the authors can use a universal p<0.05 as a significance criterion. Please consult the statistician and either downsample to the same sample size or use another appropriate analysis. Please add appropriate clarification in the Materials and Methods.
Answer
Thank you for raising an important point. We acknowledge that unequal sample sizes can affect statistical power and the interpretation of results. However, we have taken this into account: as the data does not meet normality assumption, non-parametric comparisons were conducted using the Mann-Whitney U test. This test is suitable for comparing independent groups with unequal sample sizes. Therefore, we believe that it is appropriate to maintain our current statistical approach without downsampling or altering the test type.
Minor:
3) Figure 1. Stereomicroscopic images: please add arrows or some other accents to show (i) a "spoked wheel" and (ii) the intact or disrupted integrity of the chorionic epithelium on the images.
Answer
Thank you for the note. We have corrected the Figure 1 stereomicroscopic images according to your remarks. The arrows indicate the "spoked wheel". The dotted line indicates disruption of the chorionic epithelium, and the arrowhead indicates intact chorionic epithelium (page 5).
4) Figure 1. H&E images: please ass extra column with the enlarged H&E images (and please mark the region that is enlarged); it's hard to see anything on the current images.
Answer
Thank you for the observation. We have corrected the Figure 1 H-E images according to your remarks: we have added an extra column with H-E images at 10× magnification and marked the enlarged region with an asterisk in the H-E 4× magnification images (page 5).
5) Table 1: p<0.05 is an arbitrary number that was agreed to be a significance criterion. ap = 0.047 is technically passing as a significant difference. What was the p-value for the T98G-2 mM VPA-3 mM NaDCA?
Answer
The p-value for T98G-2 mM VPA-3 mM NaDCA is 0.08.
6) Figure 2: Please add either the data points to the bar graph, or ideally, present the data as a box plot.
Answer
Thank you for your remark. Unfortunately, we are unable to add data points or represent the data as a box plot because the data represented in the graph is qualitative (invasion or no invasion), not quantitative. The percentage in the graph represents the number of invasive tumors in each of the studied groups.
7) Figure 6I: The red arrows are confusing. I'd suggest removing the arrows from "zoom-out" panels and adding the enlarged "zoom-in" of the significant regions showing the stain-positive and stain-negative cells on the enlarged images.
Answer
We have removed the red arrows and enhanced the image quality in Figures 6−8 to better distinguish between stain-positive and stain-negative cells (pages 12−16). Unfortunately, we are unable to add images at a greater magnification. The images represented in the figures are at the highest magnification possible with the microscope we have used.
8) Figure 7I: The same as Fig. 6I, please remove the red arrows from "zoom out" panels and add enlarged panels, showing the stain-positive and negative cells there.
Answer
Kindly thank you for your remark. We have removed the red arrows and enhanced the image quality in Figures 6−8 (pages 12−16). Unfortunately, we are unable to add images at a greater magnification. The images represented in the figures are at the highest magnification possible with the microscope we have used.
9) Line 451: "Kaunas" is missing in the "Lithuanian University of Health Sciences, Lithuania"
Answer
Corrected by comment.
We are grateful to the reviewer for comments, on which we have made significant improvements to the manuscript.
Sincerely,
Donatas Stakišaitis

Round 2
Reviewer 2 Report
Comments and Suggestions for Authors
The suggestions I have provided in my review of the manuscript are appropriate, as they have contributed to a clearer understanding of the methodology and results, while also enhancing the overall credibility of the study. Concerning the abstract, although this aspect was not addressed by other reviewers, I would kindly suggest that it be revised to emphasize the innovative elements and main findings of the study, and to ensure it is presented in a clear and accessible manner, without excessive technical detail.
Comments on the Quality of English LanguageAdditionally, I would kindly recommend that the manuscript be reviewed by a native English speaker or a professional language editor, in order to improve clarity, fluency, and overall linguistic accuracy.
Author Response
Review Report Form REVIEWER 2 (Round 2)
Open Review
Below are the Reviewer's comments and the Authors’ corrections, respectively.
Comments and Suggestions for Authors
The suggestions I have provided in my review of the manuscript are appropriate, as they have contributed to a clearer understanding of the methodology and results, while also enhancing the overall credibility of the study. Concerning the abstract, although this aspect was not addressed by other reviewers, I would kindly suggest that it be revised to emphasize the innovative elements and main findings of the study, and to ensure it is presented in a clear and accessible manner, without excessive technical detail.
Answer
The abstract is rewritten
Abstract: To date, there is no effective treatment for glioblastoma. The study aimed to compare the effectiveness of sodium dichloroacetate (NaDCA) or a valproic acid and NaDCA combination (VPA–NaDCA) or temozolomide (TMZ) on the U87 and T98G cell tumors on CAM, on the expression of PCNA, EZH2, and p53 in tumors on CAM, and SLC12A2, SLC12A5, SLC5A8 and CDH1 and CDH2 in cells. VPA–NaDCA and TMZ reduced the invasion of U87 and T98G tumors, as well as the expression of PCNA and EZH2 in the tumor. TMZ reduced p53 expression in tumors from both cell lines, whereas VPA–NaDCA did not affect the expression of this marker. VPA–NaDCA, but not TMZ, reduced SLC12A2 expression in T98G cells. However, VPA–NaDCA and TMZ did not affect SLC12A2 expression in U87 cells. VPA–NaDCA increased SLC5A8 expression only in U87 cells, and TMZ did not affect gene expression in either cell line. Only VPA–NaDCA increased CDH1 expression and decreased CDH2 expression in T98G cells, whereas TMZ had no effect on gene expression in the study cells. The study demonstrated that VPA–NaDCA exhibits a more effective anticancer effect than NaDCA. The data suggest that VPA–NaDCA has a more effective impact than TMZ; however, the effect of investigational medicines on carcinogenesis varies depending on the cell line. The study of the efficacy of drugs in tumors on CAM and cells demonstrates that it is essential to assess the effectiveness of treatment, which should be personalized, before administering chemotherapy.
Comments on the Quality of English Language
Additionally, I would kindly recommend that the manuscript be reviewed by a native English speaker or a professional language editor, in order to improve clarity, fluency, and overall linguistic accuracy.
Answer
We have asked the English version of the text to be corrected by the journal's language correction service.
We thank the Reviewer for comments. The text of the Abstract has been rewritten in line with the request.
The corrections have been made with Track Changer and are visible.
Sincerely,
Donatas Stakišaitis
